# Unlocking the Bioactive Potential and Exploring Novel Applications for Portuguese Endemic *Santolina impressa*

**DOI:** 10.3390/plants13141943

**Published:** 2024-07-15

**Authors:** Jorge M. Alves-Silva, Sónia Pedreiro, Mónica Zuzarte, Maria Teresa Cruz, Artur Figueirinha, Lígia Salgueiro

**Affiliations:** 1Univ Coimbra, Coimbra Institute for Clinical and Biomedical Research (iCBR), Faculty of Medicine, Azinhaga de Santa Comba, 3000-548 Coimbra, Portugal; jmasilva@fmed.uc.pt (J.M.A.-S.); mzuzarte@uc.pt (M.Z.); 2Univ Coimbra, Faculty of Pharmacy, Azinhaga de Santa Comba, 3000-548 Coimbra, Portugal; uc2007119618@student.uc.pt (S.P.); trosete@ff.uc.pt (M.T.C.); amfigueirinha@ff.uc.pt (A.F.); 3Associated Laboratory for Green Chemistry (LAQV) of the Network of Chemistry and Technology (REQUIMTE), University of Porto, 4099-002 Porto, Portugal; 4Univ Coimbra Center for Neuroscience and Cell Biology (CNC-UC), Faculty of Medicine, Rua Larga, 3004-504 Coimbra, Portugal; 5Univ Coimbra, Chemical Engineering and Renewable Resources for Sustainability (CERES), Department of Chemical Engineering, 3030-790 Coimbra, Portugal

**Keywords:** *Santolina impressa*, antioxidant, inflammation, senescence, phenolic compounds, infusion

## Abstract

The infusion of *Santolina impressa*, an endemic Portuguese plant, is traditionally used to treat various infections and disorders. This study aimed to assess its chemical profile by HPLC-DAD-ESI-MS^n^ and validate its anti-inflammatory potential. In addition, the antioxidant capacity and effects on wound healing, lipogenesis, melanogenesis, and cellular senescence, all processes in which a dysregulated inflammatory response plays a pivotal role, were unveiled. The anti-inflammatory potential was assessed in lipopolysaccharide (LPS)-stimulated macrophages, cell migration was determined using a scratch wound assay, lipogenesis was assessed on T0901317-stimulated keratinocytes and melanogenesis on 3-isobutyl-1-methylxanthine (IBMX)-activated melanocytes. Etoposide was used to induce senescence in fibroblasts. Our results point out a chemical composition predominantly characterized by dicaffeoylquinic acids and low amounts of flavonols. Regarding the infusion’s bioactive potential, an anti-inflammatory effect was evident through a decrease in nitric oxide production and inducible nitric oxide synthase and pro-interleukin-1β protein levels. Moreover, a decrease in fibroblast migration was observed, as well as an inhibition in both intracellular lipid accumulation and melanogenesis. Furthermore, the infusion decreased senescence-associated β-galactosidase activity, γH2AX nuclear accumulation and both p53 and p21 protein levels. Overall, this study confirms the traditional uses of *S. impressa* and ascribes additional properties of interest in the pharmaceutical and dermocosmetics industries.

## 1. Introduction

The global population of individuals aged 60 years old and above is projected to reach 2.1 billion by 2050 [1], posing significant challenges to public health. Indeed, the ageing process leads to a decline in several body functions and increases the risk of developing age-related disorders, such as neurodegenerative, cardiovascular, cancer and metabolic diseases [2], thus increasing morbidity and mortality. Ageing is often associated with a decrease in the endogenous antioxidant system, resulting in elevated production of reactive oxygen species (ROS) [3] which, in turn, promotes cellular senescence by activating the p16-RB signaling pathway via ERK activation. In addition, ROS production triggers a senescence-associated secretory phenotype (SASP) and inflammaging (chronic inflammation associated with ageing) by activating the NF-κB pathway [4]. Moreover, as reviewed elsewhere, both ROS and reactive nitrogen species contribute to skin alterations, including melanogenesis [5]. Indeed, it has been shown that ROS increases skin pigmentation by promoting the conversion of dihydroindole to indole quinone [6]. Furthermore, nitric oxide sensitizes melanocytes to α-melanocyte stimulating hormone (α-MSH) [7] and nitric oxide (NO) released by UV-stimulated keratinocytes stimulates melanogenesis [8].

As a result, there is growing interest in identifying natural antioxidants namely of plant origin, particularly polyphenolic-rich extracts that can decrease oxidative stress and age-related conditions [9,10,11,12]. Indeed, ROS scavengers have shown promising effects in preventing melanogenesis [13] and molecules activating the Nrf2 pathway are of particular interest, as this pathway regulates melanogenesis [14]. Moreover, plants with proven health benefits have increased economic interest and are highly valued as industrial crops.

*Santolina impressa* Hoffmanns. & Link (Asteraceae) is an aromatic endemic plant that grows in the southwest of Portugal (estuary of Sado river to cape Sines) [15,16] and has proven health benefits and potential economic value. Traditionally, the plant’s infusions are used to treat several infections and disorders [17,18,19]. Previous studies, carried out by our group, showed a medicinal potential for *S. impressa* essential oil [20]. Bearing in mind the traditional use of *S. impressa*´s infusion, we now sought to assess the bioactive potential of this widely used extract, by resorting to an all-encompassing approach aiming at increasing this species industrial value. Therefore, the aim of this study was to unveil the anti-inflammatory potential of *S. impressa* infusion, thereby confirming its traditional uses. Furthermore, to promote a greater economic interest of this species and bearing in mind the interconnection between inflammation and skin ageing, with inflammatory skin diseases representing the number one problem in dermatology, several additional properties were evaluated, including antioxidant, wound healing, depigmenting, and anti-senescent effects. Indeed, by addressing these properties an overall delay in skin ageing could be achieved. Additionally, a comprehensive chemical composition analysis was conducted using HPLC-DAD-ESI-MS^n^, which pointed out high amount of dicaffeoylquinic acids and lower of amounts of flavonols. We also demonstrated that this extract exerts anti-inflammatory effects via the NF-κB signaling pathway, thus contributing to validate its traditional uses. In addition, the extract decreased cell migration, while inhibiting lipogenesis and hyperpigmentation. Furthermore, it also exerted anti-senescent effects by decreasing senescence associated β-galactosidase, nuclear accumulation of γH2AX and by reducing the protein levels of p53 and p21. Overall, the present study promotes the industrial interest in *S. impressa* particularly for the dermocosmetic field.

## 2. Results

### 2.1. Phenolic Composition of Santolina impressa Infusion

The infusion obtained from the flowering aerial parts of *S. impressa* showed a variety of phenolic acids mainly caffeoylquinic acid derivatives, a hydroxybenzoic acid (protocatechuic acid glycoside) and myricetin and quercetin glycosides (Figure 1, Table 1).

#### 2.1.1. Phenolic Acids

Phenolic acids were tentatively identified using UV and MS data. Compound **2** showed a molecular ion [M-H]^−^ and a MS^2^ base peak at *m/z* 315. The MS^2^ fragment at *m/z* 153 was originated from the loss of 162 a.m.u. indicating the loss of a hexose unit. Thus, compound **2** was tentatively identified as protocatechuic acid hexoside. The UV spectrum profile exhibited for compound 2 is also characteristic of this specific hydroxybenzoic acid [21]. The infusion of *S. impressa* is mainly composed by caffeic acid derivatives, namely caffeoylquinic and dicaffeoylquinic acids derivatives, with ferulic and quinic acids also identified. Among the hydroxycinnamic acids, caffeic or ferulic acid derivatives present an absorbance maximum at approximately 320 nm, and a shoulder between 250 and 300 nm [22]. Among these, compound **4** exhibited a molecular ion [M-H]^−^ at *m/z* 707, with a MS^2^ base peak at *m/z* 353 indicating the presence of an adduct of a caffeoylquinic acid. The loss of a caffeoyl unit originated the MS^3^ base peak at *m/z* 191 ([M-H-353-162]^−^), and the low intensity of fragment at *m/z* 179 are characteristic of the fragmentation pattern of 5-caffeoylquinic acid [23,24]. Compound **5** showed a molecular ion [M-H]^−^ at *m/z* 353, and the MS^2^ and MS^3^ base peaks at *m/z* 191 with a low intensity peak at *m/z* 179 and 173. This fragmentation profile is identical to the compound **4** being tentatively identified as 5-caffeoylquinic acid isomer [23,24]. Compounds **10**, **11**, **12**, **13**, **14** and **15** exhibited the same molecular ion [M-H]^−^ at *m/z* 515 and MS^2^ base peak at *m/z* 353 that resulted from the loss of a caffeoyl moiety (162 a.m.u.) [25]. This fragmentation profile is typical of caffeoylquinic acid derivatives. Through the analysis of MS^3^ spectra and the elution order, compounds **10** and **11** were tentatively identified as 1,5-*O*-dicaffeoylquinic acid isomer or 3,4-*O*-dicaffeoylquinic acid isomer, as they share the same MS^3^ base peak at *m/z* 173 (from the loss of a caffeic acid unit [M-H-180]^−^) [24,26,27]. Due to the presence of the MS^3^ base peak at *m/z* 191, from the loss of another caffeoyl unit, and the high abundance of fragment at *m/z* 179, compounds **12** and **13** were tentatively identified as 3,5-*O*-dicaffeoylquinic acid or 1,3-*O*-dicaffeoylquinic acid isomers [24,26,28]. The MS^3^ spectra of compounds **14** and **15** exhibited a base peak at *m/z* 173 (loss of a caffeic acid unit) with high intensity peak at *m/z* 179 and low intensity peak at *m/z* 191. Compounds **14** and **15** were tentatively identified as 1,4-*O*-dicaffeoylquinic acid or 4,5-*O*-dicaffeoylquinic acid isomers [24,26,27,28]. Compound **6** was tentatively identified as ferulic acid due to the presence of a molecular ion [M-H]^−^ at *m/z* 193, and MS^2^ and MS^3^ base peaks at *m/z* 149 [M-H-CO_2_]^−^ [29]. Quinic acid, with a molecular ion [M-H]^−^ at *m/z* 191 (Compound **1**) was also detected in the sample [25].

#### 2.1.2. Flavonols

Flavonols were identified through their UV profile and MS fragmentation pattern [12,30]. Compound **7** exhibited a molecular ion [M-H]^−^ at *m/z* 479 and MS^2^ and MS^3^ base peaks at *m/z* 317 that resulted from the loss of 162 a.m.u indicating the presence of a hexosyl moiety linked by an oxygen to the aglycone (myricetin, MW of 318 a.m.u.). Thus, compound **7** was tentatively identified as myricetin-*O*-hexoside [21]. Compounds **8** and **9** exhibited identical mass fragmentation patterns with a molecular ion [M-H]^−^ at *m/z* 463 and MS^2^ and MS^3^ base peaks at *m/z* 301 that resulted from the loss of a hexosyl unit ([M-H-162]^−^) releasing the aglycone (quercetin, MW of 302 a.m.u.). Compounds **8** and **9** were tentatively identified as quercetin-*O*-hexoside isomers [21].

### 2.2. Antioxidant Activity of Santolina impressa Infusion

The infusion of *S. impressa* disclosed a high scavenging activity, with an IC_50_ of 25.29 ± 3.14 μg/mL and 19.00 ± 1.26 μg/mL, through DPPH and ABTS assays, respectively (Table 2).

### 2.3. Effect on Cell Viability

With the anticipation of a potential future application of *S. impressa* infusion in pharmaceutical and/or dermocosmetic products, our initial focus was to evaluate the safety profile of the infusion in several cell lines, particularly those relevant to skin health. As depicted in Figure 2, the infusion demonstrated a safe profile in macrophages at concentrations below 300 µg/mL (Figure 2A), while exhibiting complete absence of toxicity in fibroblasts (NIH/3T3), keratinocytes (HaCaT) and melanocytes (B16V) at the tested concentrations (Figure 2B–D).

### 2.4. Santolina impressa Infusion Exerts Anti-Inflammatory Potential by Modulating the NF-κB Signaling Pathway

Inflammation was mimicked resorting to the in vitro model of lipopolysaccharide (LPS)-stimulated macrophages, since LPS is a well-known agonist of Toll-like receptor 4, which triggers intracellular pro-inflammatory cascades, particularly that of NF-κB. Indeed, in macrophages treated with LPS (50 ng/mL) for 24 h, the nitrites detected in the culture medium were greatly increased when compared to control cells ([Nitrites] = 21.26 ± 9.45 vs. 0.66 ± 0.18 µM, respectively, Figure 3A). Interestingly, pretreatment with *S. impressa* infusion for 1 h before LPS stimulation, led to a significant decrease in the detected nitrites in a dose-dependent manner (IC_50_ = 132.8 µg/mL, Figure 3A). As 200 µg/mL was the first non-toxic concentration that inhibited more than 50% of nitrite release, it was selected to disclose the mechanism by which *S. impressa* exerts its anti-inflammatory potential. To achieve this goal, we addressed the protein levels of inducible nitric oxide synthase (iNOS) and pro-interleukin-1β (pro-IL-1β), two key pro-inflammatory mediators dependent of the NF-κB signaling pathway. As anticipated, cell exposure to LPS for 24 h resulted in significant upregulation of both pro-inflammatory proteins (Figure 3B–D). Interestingly, pretreatment with 200 µg/mL of *S. impressa* infusion for 1 h before 24 h of LPS exposure, significantly reduced the protein levels of both iNOS and pro-IL-1β (Figure 3B–D), thus confirming the anti-inflammatory potential of the infusion.

### 2.5. Santolina impressa Infusion Delays Cell Migration

As ageing frequently results in diminished tissue regenerative capacity [31], by enhancing this capacity one could potentially mitigate age-related chronic diseases. With this in mind, we assessed the migratory capacity of fibroblasts in the presence of *S. impressa* infusion. As shown in Figure 4, 200 µg/mL of the infusion significantly inhibited wound healing properties of fibroblasts, while a lower dose (100 µg/mL) had no effect on NIH/3T3 cells migration.

Considering that wound healing is already compromised in ageing, the dose of 100 µg/mL was selected to assess the remaining bioactivities. Nevertheless, 200 µg/mL could be explored in the setting of cancer therapies, as cell migration plays a pivotal role in cancer invasion [32].

### 2.6. Santolina impressa Infusion Decreases Lipogenesis

UV-induced skin photoaging commonly triggers increased lipid production in keratinocytes [33,34,35]. Consequently, therapies aimed at reducing lipogenesis may hold significant interest in managing this cellular event.

Keratinocytes stimulated with the LXR agonist T0901317 exhibited a notable increase in oil red O content (Figure 5A,B), confirming the activation of lipogenesis upon LXR activation and, thereby, validating this cellular mechanism. Interestingly, the presence of *S. impressa* infusion significantly reduced oil red O content (Figure 5A,B), showing that it decreases lipogenesis.

### 2.7. Santolina impressa Prevents Hyperpigmentation

Hyperpigmentation can result from inflammation, UV radiation, and ageing [36], making depigmenting compounds highly sought after in the dermocosmetics industry. Taking this into account, we used 3-isobutyl-1-methylxanthine (IBMX) as an inducer of melanogenesis, which, as anticipated, significantly increased tyrosinase activity, subsequently elevating melanin content (Figure 6A,B). Our results demonstrated that in the presence of 100 µg/mL of *S. impressa* infusion, tyrosinase activity was significantly decreased, consequently reducing melanin content and confirming depigmenting effects of the infusion.

### 2.8. Santolina impressa Exerted Anti-Senescent Activity through Modulation of p53/p21 Pathway

Bearing in mind that all the previously reported activities are linked to ageing and senescence-related loss of function, we then assessed whether *S. impressa* infusion could tackle senescence directly, thus reducing senescent cells’ accumulation, which are known to cause age-related diseases [37]. As recommended by González-Gualda [38], three complementary senescence markers were assessed, namely those related to cell cycle arrest (p53 and p21 protein levels), increased lysosomal compartment activity (senescence-associated (SA) β-galactosidase activity) and DNA damage (nuclear accumulation of yH2AX histone) [39].

Resorting to etoposide, we clearly induced senescence in NIH/3T3 fibroblasts as observed by an increase in the SA β-galactosidase activity (Figure 7A,B). Importantly, in the presence of the infusion (100 µg/mL) this feature was significantly decreased (Figure 7A,B).

To better comprehend the anti-senescent role of *S. impressa* infusion we further explored other relevant signals, particularly the nuclear accumulation of γH2AX and the protein levels of p53 and p21. As expected, etoposide alone significantly increased the phosphorylation and nuclear accumulation of γH2AX (Figure 8A,B), a clear sign of double strand DNA damage, which was further corroborated with the increased levels of both p53 and p21 in western blot analysis (Figure 8C–E). Overall, the presence of *S. impressa* infusion decreased all the tested parameters (Figure 8A–E), thus strengthening its anti-senescent potential.

## 3. Discussion

To the best of our knowledge, this is the first study reporting the chemical profile of *S. impressa* infusion. As previously mentioned, this extract has a high content of caffeoylquinic acid derivatives and a lower amount of flavonols, namely quercetin and myricetin glycosides. Other studies have previously reported the chemical composition of *S. impressa* decoction, showing that chlorogenic acid (5-caffeoylquinic acid), and dicaffeoylquinic acids, namely dicaffeoylquinic acid and 1,3-dicaffeoylquinic acid were the major phenolic compounds. Flavonoids were also detected, namely homoorientin and myricetin-3-*O*-glucoside [40]. Additionally, the phenolic composition of a hydromethanolic extract from *Santolina chamaecyparissus* showed a high abundance of cynarin (1,5-dicaffeoylquinic acid), followed by chlorogenic acid, and lower amounts of flavonoids, particularly quercetin and luteolin derivatives [41]. Gomes et al. studied the polyphenolic content of a hydroethanolic extract (50% *v*/*v*) of leaves and stalks from *Santolina semidentata* by HPLC-DAD. The analysis performed showed high concentrations of hydroxycinnamic acids followed by flavones with simple phenolic acids, ferulic acid and its derivatives and flavonols [30]. Quinic acid, protocatechuic acid hexoside, ferulic acid and dicaffeoylquinic acid isomers were herein reported for the first time in *S. impressa* and interestingly for *Santolina* genus.

Regarding the antioxidant potential of the infusion, this study is also the first report on *S. impressa*’s infusion, showing significant antioxidant activity. This activity may be attributed to the infusion´s polyphenolic content, particularly rich in chlorogenic acid derivatives. Indeed, a similar antioxidant activity was reported for decoctions obtained from the stem/leaf and capitula of *S. impressa* and related with the presence of cynarin and chlorogenic acid [40]. A higher antioxidant effect was observed for the hydromethanolic extract obtained from *S. chamaecyparissus* leaves with IC_50_ of 8.02 µg/mL [42]. Interestingly, polyphenols (such as caffeoylquinic acids) are electron-rich compounds with phenolic hydroxyl groups in a conjugated system [42,43]. Their free radical scavenging activity is related to the hydrogen donating ability of the hydroxyl group (attached to the phenyl group) by the formation of a stable phenoxyl radical through the unpaired electron delocalization [42,43]. Additionally, since caffeoylquinic acids can be oxidized in vivo to quinoids, they are also Michael acceptors exhorting their antioxidant potential via Nrf2 pathway activation (a therapeutic target in ageing related diseases) [43]. The dicaffeoylquinic acids have two cinnamic acid moieties containing more phenolic hydrogen atoms than mono-caffeoylquinic acids [44]. In this way, Liang et al., studied the antioxidant activity of caffeoylquinic and dicaffeoylquinic acids in ABTS, NO radical scavenging and ORAC assays and in Caco-2 cells [44]. The results showed a higher anti-radical activity of dicaffeoylquinic acids compared to mono-caffeoylquinic acids [44]. In accordance, the results herein obtained indicate an antioxidant and anti-inflammatory potential of *S. impressa* infusion, justifying further in vivo validations.

Skin ageing and skin disorders, like eczema, acne, atopic dermatitis and psoriasis, often involve an overproduction of pro-inflammatory cytokines and are influenced by the skin immune response [45,46,47]. Current therapies such as glucocorticoids and immunosuppressants have driven the search for new alternatives [48,49]. Our prior research demonstrated that the essential oil of *S. impressa* significantly decreased NO release in LPS-stimulated macrophages, in non-toxic concentrations, while also decreasing the protein levels of iNOS [20]. To strengthen the interest in this species, and given its traditional uses, we herein explored the anti-inflammatory effects of *S. impressa* infusion and the results obtained showed a significant decrease in NO release and in the protein levels of both iNOS and pro-IL-1β, thus confirming the anti-inflammatory potential of the infusion, the type of extract mostly used. These effects may be attributed to some of the compounds present in the extract. For example, it was reported that chlorogenic acid reduced the secretion of IL-8 and associated *il-8* gene expression induced by TNF-α [50]. In addition, 3,5-*O*-dicaffeoylquinic acid, 3,4-*O*-dicaffeoylquinic acid and 4,5-*O*-dicaffeoylquinic acid decreased the edema as well as TNF-α and IL-1β in the carrageenan-induced paw edema animal model [51]. In a different study, 3,5-dicaffeoylquinic acid and 4,5-dicaffeoylquinic acid decreased PGE_2_ and TNF-α in LPS-stimulated U-937 cells [52]. Moreover, the anti-inflammatory potential of ferulic acid [53,54,55], myricetin [56,57,58,59] and quercetin [60,61,62] have also been reported. In terms of mechanism, ferulic acid has been shown to inhibit the NRLP3 inflammasome, while myricetin inhibited NF-κB and STAT1 activation [53,56]. Given that these compounds are present in substantial quantities in *S. impressa* infusion, we suggest that the reported anti-inflammatory effect may be ascribed to the modulation of these pathways.

Ageing can impair the regenerative capacity of tissues [31], particularly when the inflammatory phase of wound healing is prolonged [63]. Considering the observed anti-inflammatory effect, we hypothesized that the infusion could contribute to the wound healing process. Unexpectedly, the infusion fails to promote wound healing; indeed, at the highest dose tested (200 µg/mL), the infusion even delayed cell migration. Contrarily, it was reported that 4,5-dicaffeoylquinic acid [64] and 5-caffeoylquinic acid [65] promote wound healing. Additionally, ferulic acid is known to promote wound healing in different models [66,67,68] and the flavonoids quercetin and myricetin, as well as their glycosidic derivatives, are also effective [65,69,70,71,72]. Considering these results, we hypothesize that when present in a complex mixture, these compounds may exhibit antagonistic interactions, thereby reducing the migratory capacity of fibroblasts.

In addition to wound healing, other skin features are highly compromised during ageing. For example, UV radiation-induced skin ageing commonly elevates lipid production in keratinocytes [33,34,35]. Our findings reveal that *S. impressa* infusion significantly mitigates lipogenesis under conditions where lipid synthesis is induced. Therefore, it holds potential for incorporation into dermocosmetic products targeting UV-induced lipogenesis in keratinocytes. The impact of the major phenolic compounds from *S. impressa* on lipogenesis has been documented. Indeed, in a model of advanced alcoholic steatohepatitis, chlorogenic acid reduced lipogenesis-related genes expression (*Srebp1* and *Acc*) [73]. Moreover, in *Caenorhabditis elegans*, 5-caffeyolquinic acid diminished lipid accumulation with concomitant decreased lipogenesis-associated genes expression, particularly, *Sbp-1* and *Daf-16* [74]. In 3T3-L1 pre-adipocytes, 3,5-dicaffeoylquinic acid reduced lipid accumulation and the protein levels of fatty acid synthase with concomitant increase in AMPK and acetyl-CoA carboxylase [75]. In the same cell type, ferulic acid reduced intracellular lipid accumulation and protein levels of C/EBP-β, C/EBP-α, PPAR-γ, and SREBP-1, while increasing that of p38MAPK, p44/42 (Erk 1/2) and phosphorylated AMPK-α [76]. Importantly, ferulic acid differentially affected lipid accumulation depending on the metabolic background. Indeed, while in the “lipid storage phase” the presence of the compound reduced lipid accumulation and altered the expression of lipogenesis-related genes, in the “lipogenic phase” the concentration of the compound required to decrease lipid content in adipocytes was increased ten-fold [77]. Both detected flavonoids have been reported to impact lipid biogenesis and lipid content in different models. Indeed, quercetin decreased lipid accumulation and the expression of genes related to adipocyte differentiation in 3T3-L1 cells [78]. Moreover, in a cell model of non-alcoholic fatty liver disease, the presence of quercetin reduced lipid accumulation by preventing de novo lipid synthesis [79]. Similarly, to what was reported for ferulic acid by Little et al. [77], quercetin affected lipid storage dependent on the metabolic state. Conversely, myricetin inhibited pre-adipocyte differentiation and increased the expression of lipolysis-related genes in 3T3-L1 pre-adipocytes [80]. In adipose tissue-derived mesenchymal stem cells, myricetin suppressed adipogenesis, by decreasing intracellular lipid accumulation [81].

While melanocytes’ amount decreases with age, atypical pigmentation remains a prevalent symptom of ageing skin. Additionally, chronic UV exposure impairs the function of fibroblasts involved in melanogenesis regulation [36] and melanocytes-senescent fibroblast communication plays a significant role in age-related pigmentation. In the present study, we showed that *S. impressa* infusion inhibited tyrosinase activity, a key player in melanin production [82], which could explain the decrease in the melanin content observed in melanocytes. Similarly to what we reported for *S. impressa*, 5-caffeyolquinic acid also decreased tyrosinase activity, as well as melanin content in α-MSH-stimulated melanocytes [83]. In addition, chlorogenic acid, as well as 3,4-; 3,5- and 4,5-dicaffeyolquinic acids inhibited melanogenesis in a dose-dependent manner [84]. Interestingly, all the dicaffeoylquinic acids present in our sample have been reported to decrease melanin content and the protein levels of tyrosinase and other melanogenesis-related proteins [85]. Ferulic acid and quercetin inhibited UVA-induced melanogenesis by modulation of the Nrf2 signaling pathway [86]. In women aged 45 to 60 years old, exhibiting skin photoaging symptoms, a 14% ferulic acid peel administered weekly for 8 weeks demonstrated a notable bleaching effect [87], highlighting the potential of phenolic compounds in mitigating skin photoaging. In α-MSH-stimulated melanocytes, the presence of quercetin decreased the activity and protein levels of tyrosinase as well as those of tyrosinase-related protein (TRP)-1 and TRP-2 [88]. Quercetin and its 4′-*O*-glycoside decreased the melanin content in melanocytes [89]. Additionally, the flavonoids myricetin 3-*O*-galactoside and quercetin 3-*O*-galactoside inhibited melanogenesis in α-MSH-stimulated melanocytes by modulating PKA and ERK1/2 activation [90,91]. In contrast, other studies demonstrated that both myricetin and quercetin increased melanin content and tyrosinase activity [92,93]. Quercetin also promoted melanogenesis in hair follicles by inducing tyrosinase activity [94]. While these studies may seem contradictory to those reported in the present study and by others, it is important to note that in the previously reported studies, both flavonoids stimulate melanogenesis in non-stimulated melanocytes. However, in activated melanocytes, both compounds exhibit an inhibitory effect on melanogenesis, which may explain the seemingly conflicting results. Overall, the results achieved in our study suggest that the presence of phenolic compounds, including caffeoylquinic acids and flavonoids, are responsible for the melanogenesis inhibition triggered by *S. impressa* in activated melanocytes.

Another relevant skin ageing hallmark is senescence [95,96]. Indeed, studies have demonstrated that in fibroblasts UVB exposition leads to DNA damage, inducing cell cycle arrest and expression of senescence markers. These markers include senescence-associated β-galactosidase activity and activation of p16, p21, and p53 [95]. Our findings indicate that *S. impressa* infusion reduces the activity of senescence-associated β-galactosidase, suggesting its potential as an anti-senescent agent. Considering this, we delved deeper into other senescence markers, as a comprehensive assessment of anti-senescent therapeutics should address three different markers [38]. We assessed γH2AX accumulation as it serves as a response to damaged DNA, inducing cell cycle arrest and consequently cellular senescence [97]. Additionally, as discussed elsewhere only repair-resistant double-strand breaks lead to cellular senescence causing the formation of a γH2AX subtype called persistent γH2AX [98]. Interestingly, our results demonstrate that the infusion is able to prevent the nuclear accumulation of γH2AX, thus suggesting that *S. impressa* could prevent double strand DNA damage, which in turn prevents cell cycle arrest and consequent cellular senescence by inhibiting the p53/p21 axis. In accordance with this, both p53 and p21 protein levels were decreased in the presence of the infusion, thus reinforcing its anti-senescent potential. Previous studies have assessed the anti-senescent potential of caffeoylquinic and ferulic acids, as well as myricetin and quercetin. The presence of 3,5-dicaffeoylquinic acid ameliorated spatial learning and memory in a senescence-accelerated-prone-mice model [99], thus suggesting that this compound might have anti-senescent properties. Ferulic acid inhibited cell senescence induced by whole body radiation bone marrow damage [100]. In UVA-treated human dermal fibroblasts, pretreatment with ferulic acid prevented cell cycle arrest and concomitant decrease in mRNA levels encoding p21. Furthermore, it decreased senescent cells and mRNA encoding p16, MMP-1 and MMP-3 [101]. Myricetin decreased SA β-galactosidase activity, p21 and p16 protein levels and IL-6 and IL-8 secretion in hydrogen peroxide-induced cellular senescence in nucleus pulposus cells [102] and in nucleus pulposus-derived mesenchymal stem cells [103]. Indeed, various studies have documented senolytic properties for quercetin. For instance, in pre-adipocytes and mature adipocytes treated with hydrogen peroxide, the presence of quercetin decreased SA β-galactosidase activity, the protein levels of p21 and pro-inflammatory cytokines secretion [104]. In human skin irradiated with UV, quercetin prevented skin photoaging by modulating several signaling pathways, leading to decreased MMP production and inhibition of collagen degradation [105]. Furthermore, in primary keratinocytes, quercetin prevented NF-κB nuclear translocation after UV treatment, thus reducing the expression of IL-1β, IL-6, IL-8 and TNF-α [106].

In summary, our findings contribute to confirm some of the traditional uses ascribed to the infusion of *S. impressa* by demonstrating strong anti-inflammatory effects via NF-κB modulation. In addition, several other biologically relevant properties in the context of skin ageing were disclosed, particularly lipogenesis and melanogenesis inhibition, as well as anti-senescent effects. Overall, the present study demonstrates that *S. impressa* has the potential to counteract various hallmarks of ageing, particularly those related to skin ageing, which in addition to the infusion’s high yield, could be applied in the development of innovative dermocosmetic formulations for the management of various skin disorders associated with aged skin.

## 4. Materials and Methods

### 4.1. Material Collection and Santolina impressa’s Infusion Preparation

Aerial flowering parts of *Santolina impressa* were randomly collected during full blossom (June 2022) from populations occurring throughout a coastal area in Setúbal region (Tróia Peninsula (38°29′17.3″ N, 8°54′37.5″ W) and Comporta Beach (38°22′53.0″ N 8°48′11.5″ W). A voucher specimen was stored in the herbarium of the Faculty of Pharmacy of the University of Coimbra, with the assessment number LS 232. Authenticity of the species was verified by Jorge Paiva, taxonomist at the University of Coimbra. The samples were air dried at 25 °C for 2 weeks, and then milled (knife mill KSM 2) and sieved (60-mesh sieve). The infusion was prepared by the addition of 1 L of previously boiled water to 10 g of the sieved plant. After 30 min, the extract was filtered, concentrated using a Rotavapor (R-114), frozen and lyophilized using a FTS System type EZDRY. The sample was then stored at −20 °C in the dark until use. The extraction yield was 23.28% (mg/g of dried plant).

### 4.2. Chemical Characterization by HPLC-DAD-ESI-MSn

The chemical composition of the infusion was determined by high-performance liquid chromatography equipped with a diode array detector (DAD) (Finnigan Surveyor, THERMO, Waltham, MA, USA) and a linear ion trap mass spectrometer (LIT-MS) (LTQ XL, THERMO, Walthman, MA, USA). 20 μL of the sample (5 mg/mL) was injected in a Waters Spherisorb ODS2 C18 column (150 × 2.1 mm and 3 μm particle size) and eluted at 20 °C with binary mixture of 2% (*v*/*v*) aqueous formic acid (solvent A) and acetonitrile (solvent B) at a flow rate of 200 μL/min. The gradient used was 5–50% (*v*/*v*) of B for 60 min. The chromatograms were recorded at wavelengths of 280 and 320 nm. The MS detector was operated at negative electrospray ion mode using Helium as collision gas (collision energy of 35%), Nitrogen as the nebulizing gas (35 arbitrary units) and as auxiliary gas (20 arbitrary units). The capillary temperature and voltage were 275 °C and −35.00 V, respectively, and the source voltage was 5.00 kV.

### 4.3. Antioxidant Activity

The scavenging activity of the infusion was assessed by the 2,2-diphenyl-1-picrylhydrazyl (DPPH) radical scavenging assay [9] and 2,2′-azino-bis-(3-ethylbenzothiazoline-6-sulfonate) (ABTS) radical scavenging assay [9]. All measurements were performed in three independent experiments carried out in duplicate. The scavenging activity was expressed as IC_50_ and Trolox-Equivalents (TE).

### 4.4. Cell Culture

The cell lines RAW 264.7 (murine macrophage cell line) and NIH/3T3 (embryonic mouse fibroblast cell line) were obtained from the American Type Culture Collection (ATCC TIB-71 and ATCC CRL-1658), while the cell lines HaCaT (immortalized human keratinocyte cell line, CLS 3004993) and B-16V (murine melanoma cell line, DSMZ ACC-370) were obtained from Cell Line Services and German Collection of Microorganisms and Cell Cultures, respectively. All cell lines were cultured following previously described methods [107,108].

### 4.5. Effect on Cell Viability

The effect of varying concentrations of the infusion on macrophages, fibroblasts, keratinocytes and melanocytes viability was assessed using the resazurin (Alamar Blue) reduction assay. Briefly, RAW 264.7 macrophages (600,000 cells/mL), NIH/3T3 fibroblasts (50,000 cells/mL), HaCaT keratinocytes (100,000 cells/mL) and B-16V melanocytes (60,000 cells/mL) were plated in 48-well plates. Following a 24 h exposure to the infusion (600–100 µg/mL for B16V, HaCaT and 3T3 cells; 400–50 μg/mL for RAW 264.7), cell viability was measured according to established methods [109].

### 4.6. Anti-Inflammatory Potential

The infusion’s capacity to reduce nitrites production in lipopolysaccharide (LPS)-stimulated macrophages was evaluated following the methods detailed in previous studies conducted by our team [20,110]. The protein levels of inducible nitric oxide synthase (iNOS) and pro-interleukin-1β (pro-IL-1β) were assessed by western blot analysis, as previously described by Alves-Silva et al. [12].

### 4.7. Cell Migration

The scratch wound assay was selected to assess the effect of the infusion on cell migration, according to that reported by Martinotti et al. [111] with slight modifications. Briefly, following the scratch induction on cultured NIH/3T3 fibroblasts, the capacity of the infusion (200 and 100 µg/mL) to modulate cell migration was assessed. After image acquisition, open wounds were quantified resorting to an ImageJ/Fiji plugin [112].

### 4.8. Inhibition of Lipogenesis

T0901317, an activator of the liver X receptor, was used to induce lipogenesis in HaCaT keratinocytes as described by Francisco et al. [113]. The effect of the pretreatment with the infusion (100 µg/mL) on lipogenesis was determined resorting to a previously reported method [114].

### 4.9. Depigmenting Effect

The depigmenting activity was determined resorting to the methodology previously described [108]. B-16V cells (600,000 cells/mL) were seeded in 6-well plates and allowed to grow for 24 h. Subsequently, cells were cultured for 48 h, in medium alone, or treated with 3-isobutyl-1-methylxanthine (IBMX, 200 µM), in the presence or absence of the infusion (100 µg/mL). At the end of the incubation period, cells were collected and prepared for melanin content and tyrosinase activity as previously reported [114].

### 4.10. Anti-Senescence Potential

Senescence was assessed by employing the senescence inducer etoposide, following the methodology outlined in previous studies [108]. Cellular senescence was induced using etoposide in NIH/3T3 cells for 24 h. Following this, cells were maintained in culture medium with or without the infusion (100 µg/mL). Subsequently, cells were collected for senescence-associated β-galactosidase, γH2AX and western blot analysis of p21 and p53 proteins, as reported in a previous study carried out in our laboratory [114].

### 4.11. Statistical Analysis

Three independent experiments, performed in duplicate, were carried out. The results are presented as mean ± SEM (standard error of the mean) values. Statistical significance was evaluated using either one-way analysis of variance (ANOVA) or Mann–Whitney test, followed by suitable post-hoc analysis using GraphPad Prism 9.3.0 software. *p* values below 0.05 were considered statistically significant.

## 5. Conclusions

Having in mind the traditional uses ascribed to *S. impressa* infusion, in the present study, we unveil potent NF-κB mediated anti-inflammatory effects. Furthermore, recognizing the significance of healthy skin ageing in preventing age-related skin disorders, we explored multiple properties in which inflammation plays a relevant role and with potential industrial applications in dermocosmetic formulations. Our findings demonstrate that the infusion exhibits robust antioxidant activity, likely attributed to modulation of the Nrf2 signaling pathway. Additionally, it effectively inhibits lipogenesis, melanogenesis, and displays anti-senescent properties. Overall, our study underscores the considerable potential of *S. impressa* for the pharmaceutical, cosmetic and food industries, owing to its diverse properties and high phenolic compound content.

## Figures and Tables

**Figure 1 plants-13-01943-f001:**
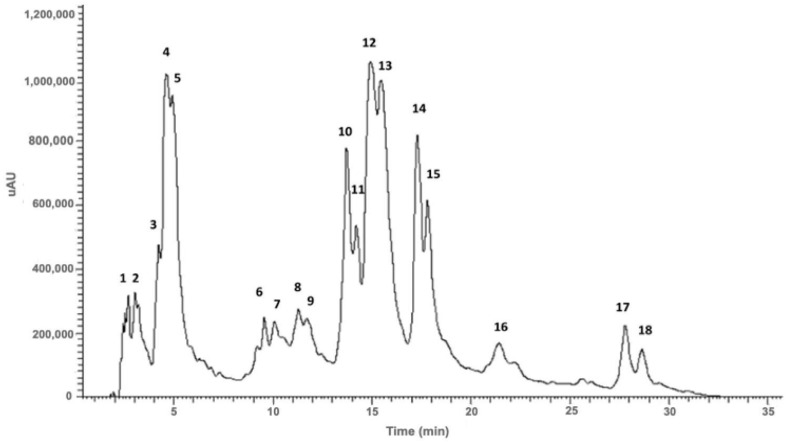
Total scan (230–600 nm) of the HPLC-DAD-ESI-MS^n^ chromatogram (0–45 min) of *S. impressa* infusion.

**Figure 2 plants-13-01943-f002:**
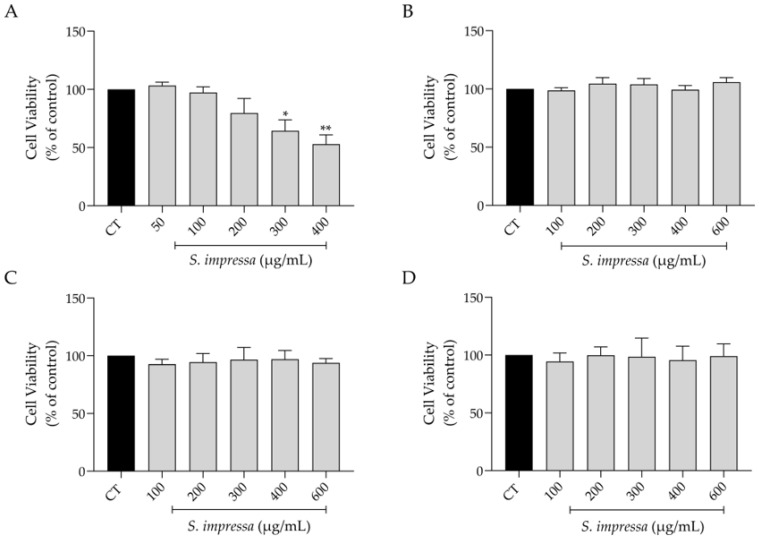
Safety profile of the infusion on macrophages (**A**), fibroblasts (**B**), keratinocytes (**C**) and melanocytes (**D**). * *p* < 0.05 and ** *p* < 0.01, compared to control (CT); one-way ANOVA followed by Dunnett’s multiple comparison test.

**Figure 3 plants-13-01943-f003:**
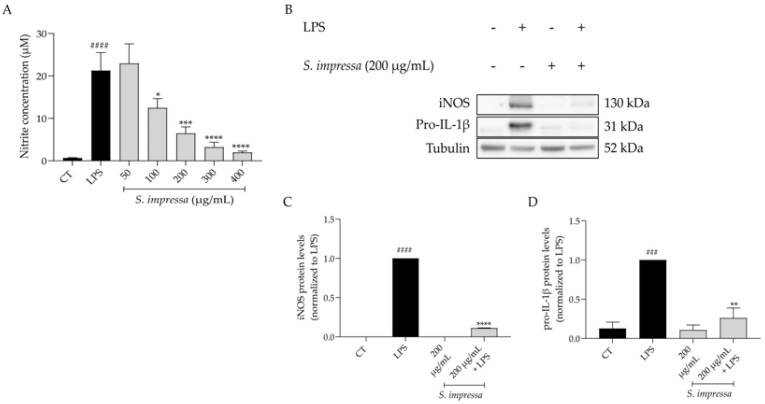
Anti-inflammatory potential of the infusion on lipopolysaccharide (LPS)-stimulated macrophages. Pretreatment of the cells with the infusion decreased nitrites production (**A**) and the protein levels of iNOS (**B**,**C**) and pro-IL-1β (**B**,**D**). Tubulin was used as loading control and values were normalized to LPS. n = 3. ^###^ *p* < 0.001 and ^####^ *p* < 0.0001, compared to control (CT); * *p* < 0.05, ** *p* < 0.01, *** *p* < 0.001 and **** *p* < 0.0001, compared to LPS; one-way ANOVA followed by Tukey’s multiple comparison test.

**Figure 4 plants-13-01943-f004:**
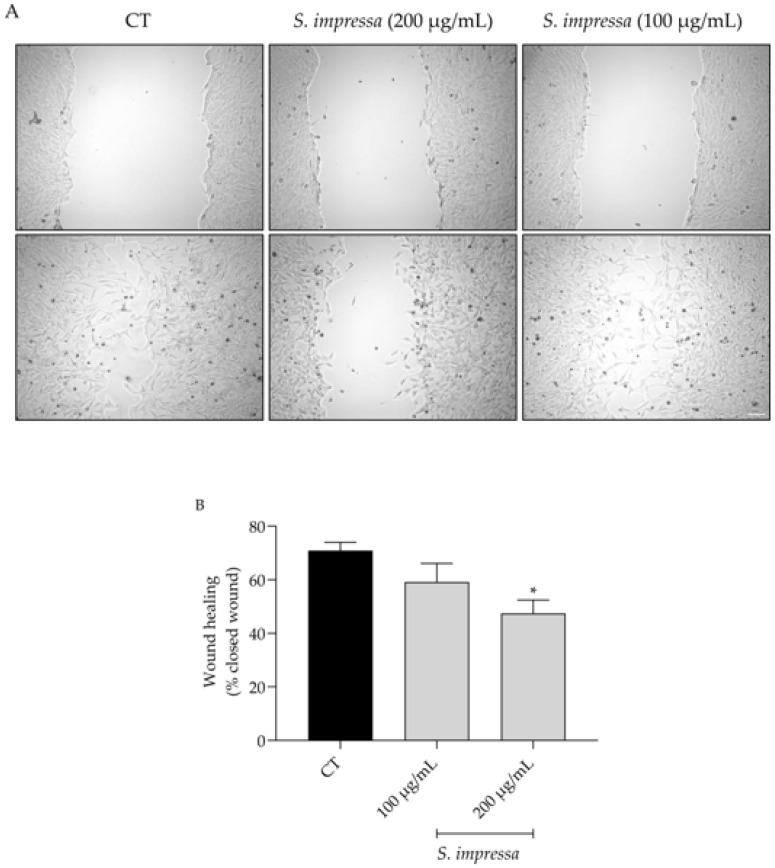
Wound healing potential of the infusion in fibroblasts. The presence of *S. impressa* infusion (200 and 100 µg/mL) decreased cell migration capacity 18 h after wound induction (**A**,**B**). * *p* < 0.05, compared to control (CT); one-way ANOVA followed by Dunnet’s multiple comparison test. Scale bar: 100 µm.

**Figure 5 plants-13-01943-f005:**
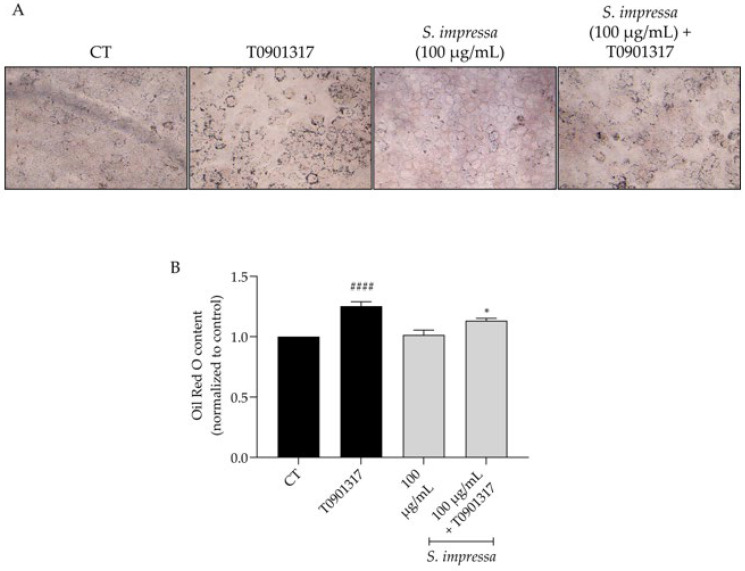
Effect of the infusion on lipogenesis in keratinocytes. Pretreatment with *S. impressa* infusion (100 µg/mL) led to a decrease in neutral lipid content, as measured by Oil Red O staining (**A**,**B**). ^####^ *p* < 0.0001, compared to control (CT) and * *p* < 0.05, compared to T0901317; one-way ANOVA followed by Tukey’s multiple comparison test. Scale bar: 50 µm.

**Figure 6 plants-13-01943-f006:**
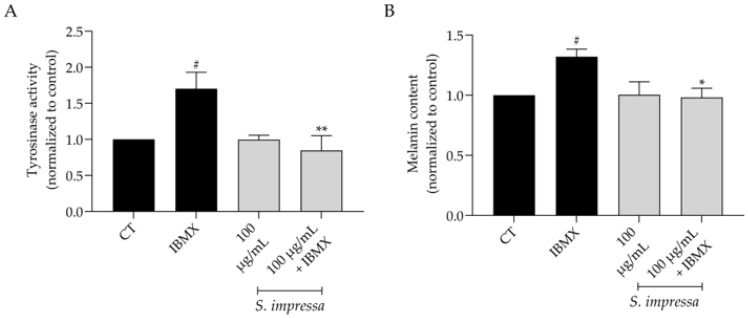
Effect of the infusion on melanogenesis in 3-isobutyl-1-methylxanthine (IBMX)-stimulated melanocytes. The presence of *S. impressa* infusion (100 µg/mL) reduced tyrosinase activity (**A**) and melanin content (**B**). ^#^ *p* < 0.05, compared to control (CT), * *p* < 0.05, *** p <* 0.01 compared to IBMX; one-way ANOVA, followed by Tukey’s multiple comparison test.

**Figure 7 plants-13-01943-f007:**
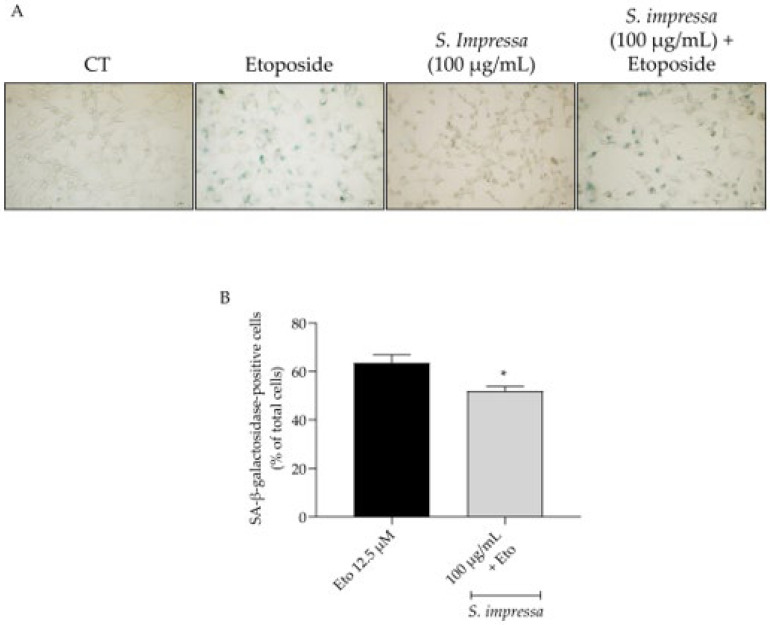
Effect of the infusion on senescence associated β-galactosidase activity. The addition of *S. impressa* infusion (100 µg/mL) in the recovery phase led to a decrease in the percentage of X-galactose-positive cells (**A**,**B**). * *p* < 0.05, compared to Eto 12.5 µM; Mann–Whitney test. Scale bar: 50 µm.

**Figure 8 plants-13-01943-f008:**
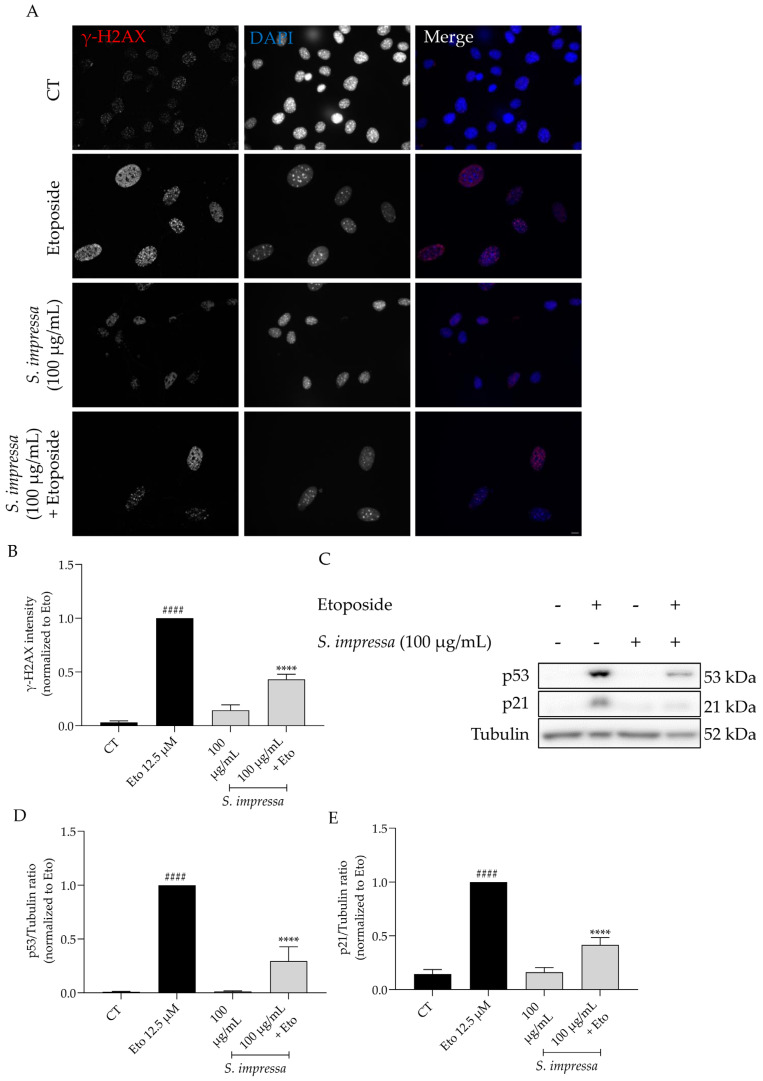
Effect of the infusion on the nuclear accumulation of the phosphorylated form of H2AX (γH2AX) and p53/p21 signaling pathway. The addition of *S. impressa* infusion in the recovery phase led to a decrease in the nuclear accumulation of γH2AX (**A**,**B**) and in the protein levels of p53 (**C**,**D**) and p21 (**C**,**E**). Tubulin was used as loading control and values were normalized to etoposide. n = 3. Six images were analyzed per condition. ^####^ *p* < 0.0001, compared to control (CT); **** *p* < 0.0001, compared to Eto 12.5 µM; one-way ANOVA, followed by Tukey’s multiple comparison test. Scale bar: 10 µm.

**Table 1 plants-13-01943-t001:** Chemical characterization of *S. impressa* infusion by HPLC-DAD-ESI-MS^n^.

Peak	Partial Identification	*R_t_* (min)	λ_max._ (nm)	[M-H]^−^	MS^2^	MS^3^
**1**	Quinic acid	2.62	242, 258	191	[191]: 173 (30), 111 (100)	-
**2**	Protocatechuic acid hexoside	3.05	242, 291	315	[315]: 315 (100), 163 (10), 153 (80)	-
**3**	Unknown	4.23	263, 299, 324 sh	255 (100)	[255]: 255 (5), 237 (20), 212 (5), 211 (100)	[255 211]: 211 (100), 165 (75), 75 (25)
**4**	5-*O*-caffeoylquinic acid	4.62	259, 299, 312 sh, 329	707 *	[707]: 353 (100), 295 (2)	[707 353]: 191 (100), 179 (10)
**5**	5-*O*-caffeoylquinic acid	4.93	255, 299, 313, 329	353	[353]: 191 (100), 179 (10), 135 (3)	[353 191]: 191 (100), 173 (25), 127 (20)
**6**	Ferulic acid	9.57	242, 291, 319	193	[193]: 149 (100)	[193 149]: 161 (50), 149 (100), 133 (75)
**7**	Myricetin-*O*-hexoside	10.09	242, 281, 324 sh	479	[479]: 317 (100), 281 (2)	[479 317]: 317 (100), 255 (2)
**8**	Quercetin-*O*-hexoside isomer	11.29	242, 282, 330 sh	463	[463]: 301 (100), 287 (10)	[463 301]: 301 (100), 283 (4)
**9**	Quercetin-*O*-hexoside isomer	11.78	242, 282, 330 sh	463	[463]: 301 (100), 463 (20)	[463 301]: 301 (100), 283 (4)
**10**	1,5-*O*-Dicaffeoylquinic acid or 3,4-*O*-Dicaffeoylquinic acid	13.72	254, 299, 326	515	[515]: 353 (100), 335 (10), 173 (10)	[515 353]: 191 (40), 179 (70), 173 (100), 135 (5)
**11**	1,5-*O*-Dicaffeoylquinic acid or 3,4-*O*Dicaffeoylquinic acid	14.17	253, 299, 328	515	[515]: 353 (100), 335 (10), 179 (5)	[515 353]: 191 (40), 179 (55), 173 (100), 135 (10)
**12**	1,3-*O*-Dicaffeoylquinic acid or 3,5-*O*-Dicaffeoylquinic acid	14.92	260, 299, 333	515	[515]: 353 (100), 179 (2)	[515 353]: 191 (100), 179 (50), 173 (10), 135 (8)
**13**	3,5-O-Dicaffeoylquinic acid or 1,3-*O*-Dicaffeoylquinic acid	15.48	253, 299, 324	515	[515]: 353 (100), 191 (2)	[515 353]: 191 (100), 179 (50), 173 (5), 135 (5)
**14**	4,5-*O*-Dicaffeoylquinic acid or 1,4-*O*-Dicaffeoylquinic acid	17.28	253, 299, 324	515	[515]: 353 (100), 299 (15)	[515 353]: 191 (30), 179 (50), 173 (100), 135 (2)
**15**	4,5-*O*-Dicaffeoylquinic acid or 1,4-*O*-Dicaffeoylquinic acid	17.80	253, 300, 326	515	[515]: 353 (100), 299 (15)	[515 353]: 191 (28), 179 (60), 173 (100), 135 (10)
**16**	Unknown	21.23	220, 287, 331 sh	725	[725]: 707 (10), 563 (100),	[725 563]: 389 (100)
**17**	Unknown	27.81	250, 330	451	[451]: 451 (30), 407 (100), 261 (20), 246 (15), 179 (40)	-
**18**	Unknown	28.70	251, 330	535	[535]: 493 (50), 491 (75), 467 (100), 447 (65), 405 (60)	-

* adduct; sh—shoulder; λ max. (maximum wavelength in UV-vis spectrum).

**Table 2 plants-13-01943-t002:** Antioxidant activity of *S. impressa* infusion assessed through DPPH and ABTS methods.

	IC_50_ (μg/mL) ^a^	TE (μmol/L) ^b^
DPPH	25.29 ± 3.14	4.13 ± 0.22
ABTS	19.00 ± 1.26	3.75 ± 0.31

^a^ Results are expressed as mean ± SD (three independent experiments, performed in duplicate). ^b^ Trolox Equivalent.

## Data Availability

Data will be made available upon request.

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
