# Peer review of "Unlocking the Bioactive Potential and Exploring Novel Applications for Portuguese Endemic Santolina impressa"

_plants, 2024, doi:10.3390/plants13141943_

Round 1

Reviewer 1 Report

Comments and Suggestions for Authors

I have very minor comments on this manuscript 

Figure 1. the graphical quality of the HPLC- trace is a bit low. You get pixelation of the steep parts of the peaks, which makes it harder to distinguish shoulders

Figure2. e.g. Usual the smallest values on an abscissa is to the left, which take some adjusting to (even for a spectroscopist like me)

Legend figure 5: the meaning of #### should be clearer (maybe even ####: ), same for figure 8.

Line 453. infusion -> infusions

LINR 459 MS Detector -> The MS detector

Line 523 (section 4.11) is not completely unambiguous.

Otherwise very decent manuscript

Comments on the Quality of English Language

The quality of Your English is high. I can see that You have already cleaned up most of it.  

Line 98. One hydroxi. that should be hydroxy

Author Response

• Figure 1. the graphical quality of the HPLC- trace is a bit low. You get pixelation of the steep parts of the peaks, which makes it harder to distinguish shoulders We thank the reviewers for the suggestion and have provided a new figure with improved quality. • Figure2. e.g. Usual the smallest values on an abscissa is to the left, which take some adjusting to (even for a spectroscopist like me) We acknowledge the reviewer´s concern and modified all the graphs accordingly, being the lowest value shown on the left of the abscissa. • Legend figure 5: the meaning of #### should be clearer (maybe even ####: ), same for figure 8. We thank the reviewer for the suggestion; however the hashtags (####) refers to p values similarly to the asterisks (***). We used hashtags when comparing a certain condition to the control and asterisks for other comparisons, to avoid confusion. The separation between the hashtags and the p, by a colon, in our opinion is confusing so, to enable a more clear reading, the hashtags were written in superscript. • Line 453. infusion -> infusions We acknowledge the suggestion made by the reviewer. The correction performed was to “the infusion” as only one infusion was analysed. • LINR 459 MS Detector -> The MS detector We agree with the suggestion and corrected the sentence accordingly. • Line 523 (section 4.11) is not completely unambiguous. Indeed the sentence is quite confusing. We have corrected it to ‘Three independent experiments, performed in duplicate, were carried out.’ • Line 98. One hydroxi. that should be hydroxy We thank the reviewer for pointing out this typo. It was corrected as suggested.

Reviewer 2 Report

Comments and Suggestions for Authors

Manuscript Title: Unlocking the bioactive potential and exploring novel applications for Portuguese endemic Santolina impressa

General comments:

This is an interesting manuscript concerned with the chemical profiling of traditionally used plant Santolina impressa. by HPLC-PDA-ESI- 18 MSn and validate its anti-inflammatory potential. However, it needs modifications before it can be accepted for publication. Below, please, see my suggestions and comments for each section of the manuscript.   

Keywords: I believe that there are better keywords in the manuscript than " skin; ageing". In my opinion, these terms are too general and do not reflect the article's main points.

ABSTRACT

1.     Write the full form of NO, iNOS and pro-IL-1β.

Introduction:

1. (Page no. 2, Line no. 59): Verify the scientific name of the plant in “The world flora online”(https://www.worldfloraonline.org/search?query=Santolina+impressa+Hoffmanns+%26+Link). It should be ‘Santolina impressa Hoffmanns. & Link’ instead ofSantolina impressa Hoffmanns & Link”

Results

1.     (Page no. 4, Line no. 139): Mention the group of compounds (phenolic acids and flavonoids) in Table 1

Discussion

1.     The author should demonstrate the correlation between this work and skin disorders. Did the author utilise skin cancer cells in this experimental study to demonstrate how the extract prevents skin diseases? Please provide a comprehensive description of the findings.

Material and methods

1.     I suggest that give a photograph of the plant and the part from which the extract is taken.

2.      (Page 15; line- 472; 4.4. Cell culture) Specify the cell lines used in these examples, such as RAW 264.7, NIH/3T3, HaCaT, and B-16V, and indicate the sort of cells they represent, such as skin and breast cancer. What is utilised for for control (CT)?

3.     (Page 15; line- 479; 4.5. Cell culture) The method used to determine the effect on cell viability was the MTT assay.?

4.     (Page 15; line- 494; 4.7. Cell migration) Scratch wound assay was carried out which cell line? Mention the cell line.

Comments on the Quality of English Language

Requires enhancement

Author Response

  • Keywords: I believe that there are better keywords in the manuscript than " skin; ageing". In my opinion, these terms are too general and do not reflect the article's main points.

We thank the reviewer for the insight. The keywords ‘phenolic compounds’ and ‘infusion’ were included to better reflect the content of the study.

  • Abstract - Write the full form of NO, iNOS and pro-IL-1β.

Thank you for the suggestion. The full named of NO, iNOS and pro-IL-1β were added when first mentioned in both the abstract and remaining manuscript.

  • Introduction - (Page no. 2, Line no. 59): Verify the scientific name of the plant in “The world flora online”

(https://www.worldfloraonline.org/search?query=Santolina+impressa+Hoffmanns+%26+Link). It should be ‘Santolina impressa Hoffmanns. & Link’ instead of “Santolina impressa Hoffmanns & Link”

We thank the reviewer for pointing out the error on the authorship of the studied plant. As suggested, it was corrected.

  • Results - (Page no. 4, Line no. 139): Mention the group of compounds (phenolic acids and flavonoids) in Table 1

We acknowledge the reviewer´s suggestion; however, in the table other classes of compounds are also present besides phenolic compounds and flavonoids. Including all compound classes would be quite confusing for the reader.

  • Discussion - The author should demonstrate the correlation between this work and skin disorders. Did the author utilise skin cancer cells in this experimental study to demonstrate how the extract prevents skin diseases? Please provide a comprehensive description of the findings.

We thank the reviewer for the comment and suggestion. We would like to highlight that the present study does not address cancer nor any other specific disease. Indeed, the main objective of the study was to assess the anti-inflammatory effect of the infusion. Furthermore, bearing in mind that inflammation is one of the major hallmarks of skin ageing, we sought to further explore the potential of the infusion on other features associated with skin ageing. Therefore, several cell lines were used to assess the effect of the infusion on specific ageing hallmarks, mainly those with direct effect on the skin. To avoid misleading, we have provided a more detailed explanation of this in both the introduction and discussion sections.

  • Material and methods – I suggest that give a photograph of the plant and the part from which the extract is taken.

The infusion was obtained from flowering aerial parts of the plant as referred in section 4.1. Unfortunately, we do not have a picture of the plant.

  • Material and methods – (Page 15; line- 472; 4.4. Cell culture) Specify the cell lines used in these examples, such as RAW 264.7, NIH/3T3, HaCaT, and B-16V, and indicate the sort of cells they represent, such as skin and breast cancer. What is utilised for for control (CT)?

We appreciate the reviewer´s suggestion. As recommended, the origin of the cell lines was added in section 4.4. As controls, all cells (RAW 264.7, NIH/3T3, HaCaT and B-16V) were cultured in culture medium alone without any kind of stimulus in the respective assay.

  • Material and methods – (Page 15; line- 479; 4.5. Cell culture) The method used to determine the effect on cell viability was the MTT assay?

In our study, we used the resazurin reduction assay to assess the effect of the infusion on cell viability. The name “Alamar blue” was added in the method description to make it clearer for the reader.

  • Material and methods – (Page 15; line- 494; 4.7. Cell migration) Scratch wound assay was carried out which cell line? Mention the cell line.

We thank the reviewer for the comment. The scratch wound assay was carried out in NIH/3T3 fibroblasts, to improve clarity NIH/3T3 was added in the methods description.

  • English requires enhancement

As suggested, an overall English improvement was carried out.